

# Segmentation of polarimetric radar imagery using statistical texture

Adrien Guyot[1,2], Jordan P. Brook[2], Alain Protat[1], Kathryn Turner[2], Joshua Soderholm[1], Nicholas F. McCarthy [3], Hamish McGowan[2]

[1] Australian Bureau of Meteorology, Melbourne, Australia

[2] Atmospheric Observations Research Group, The University of Queensland, Brisbane, Australia

[3] Country Fire Authority Development Team, Melbourne, Australia

*Correspondence to*: Adrien Guyot (adrien.guyot@bom.gov.au)

**Abstract.** Weather radars are increasingly being used to study the interaction between wildfires and the atmosphere, owing to the enhanced spatio-temporal resolution of radar data compared to conventional measurements, such as satellite imagery and in-situ sensing. An important requirement for the continued proliferation of radar data for this application is the automatic identification of fire-generated particle returns (pyrometeors) from a scene containing a diverse range of echo sources, including clear air, ground and sea clutter, and precipitation. The classification of such particles is a challenging problem for

common image segmentation approaches (e.g. fuzzy logic or unsupervised machine learning) due to the strong overlap in radar variable distributions between each echo type. Here, we propose the following two-step method to address these challenges: 1) the introduction of secondary, texture-based fields, calculated using statistical properties of Gray Level Co-occurrence Matrices (GLCM), and 2) a Gaussian Mixture Model (GMM), used to classify echo sources by combining radar variables with texture-based fields from 1). Importantly, we retain all information from the original measurements by performing calculations

in the radar's native spherical coordinate system and introduce a range-varying window methodology for our GLCM calculations to avoid range-dependent biases. We show that our method can accurately classify pyrometeors' plumes, clear air, sea clutter, and precipitation using radar data from recent wildfire events in Australia and find that the contrast of the radar correlation coefficient, is the most skilful variable for the classification. The technique we propose enables the automated detection of pyrometeors' plumes from operational weather radar networks, which may be used by fire agencies for emergency

management purposes, or by scientists for case study analyses or historical event identification.



## 1 Introduction

The ability to analyse large wildfire (referred to as "bushfire" in Australia) behaviour in real-time remains one of the biggest challenges in wildfire incident and risk management. Physical processes happen at various spatio-temporal scales, from the smaller scale of fuel consumption, heat, moisture and pyrogenic emissions to larger scale vortices, downdrafts developing in the smoke plume column and associated clouds. Large wildfires are often topped by pyrocumulus or pyrocumulonimbus

(pyroCb) clouds. The dynamics and microphysics of these clouds usually evolve very rapidly, including the formation of strong updrafts and downdrafts and their associated hazards. Pyrogenic smoke plumes and clouds facilitate the transport of embers that could lit new fires when landing, and generate lightning that could ignite new fires in the case of pyroCb. Yet, our ability to observe wildfire behaviour with the high temporal resolution provided by satellite-based passive sensing technologies remains limited because pyrogenic particles often obscure the fire ground and lower levels.


To address the spatial intelligence gap, following an earlier work of Duff et al. (2018), Lareau et al. (2022) proposed using weather radar when available, to indirectly track fire progression using radar reflectivity. Weather radars can observe ash and large debris emitted by wildfires and transported aloft; this range of wildfire-borne scatterers producing weather radar echoes has been denoted as "pyrometeors" (McCarthy et al., 2018; Kingsmill et al., 2022). Conversely, "wildfire smoke" encompasses

all wildfire-borne particles, including both pyrometeors and aerosols of smaller sizes. Lareau et al. (2022) developed an algorithm to derive fire perimeters based on real-time radar reflectivity maximas. This method was tested with US Next Generation Weather Radar data for two large wildfires that occurred in Northern California. The authors showed this would benefit from being tested and applied to several large wildfires within operational weather radar coverage. More generally, weather radar remains an under-utilised observational tool despite meeting the required criteria for wildfire tracking at a high

temporal resolution (typically 5 min for operational weather radars) and high spatial resolution (from 50 to 1000 m depending on the radar distance to the fireground and radar characteristics).

The first step before applying an algorithm, such as the one proposed by Lareau et al. (2022), is to identify and segment pyrometeors' plumes and associated clouds within a weather radar volume. Often, during a wildfire event, a range of weather

radar signal returns is present within a plan position indicator (PPI) scan, in addition to the pyrometeors' plume, such as: ground, clear air, and sea clutter, precipitation, insects or biological returns. Figure 1 shows an example of such a complex scene for wildfire pyrometeors' plumes near Sydney during the 2019/2020 Black Summer wildfires. Several pyrometeors' plumes can be seen extending from the fire area in the West towards the East and stretching over more than 100 km. In that imagery, clear air returns are clearly visible within a 50 km radius from the weather radar location. While a clutter removal

algorithm has been implemented to the data (Gabella and Notarprietro, 2002), some ground clutter is likely remaining due to anomalous propagation conditions occurring on that day, similarly to what was observed in Melbourne under similar conditions



(Guyot et al., 2021). Sea clutter is also present to the South and the East of the radar (blue coloured box in Figure 1). One can also distinguish a few showers in the southwest. Extracting the weather radar returns from pyrometeors only is particularly difficult when the pyrometeors' plume is within the 50 km radius of the radar, as clear air, and pyrometeors' echoes and

intertwined. Conversely, classification of precipitation is more straightforward since its polarimetric signatures have unique characteristics.

Fuzzy logic or unsupervised clustering-based approaches based on polarimetric radar variables are commonly used for weather radar echo classification. For these methods to be effective, each radar echo class needs to occupy a distinct area in the multi-

dimensional space defined by all the input parameters with as little overlap as possible, so that robust membership functions can be established (in the case of fuzzy logic approaches) or synthetic models can be developed (in the case of unsupervised learning such as a Gaussian Mixture Model). In the example shown in Figure 1, a classification based on reflectivity, correlation coefficient and differential reflectivity seems very difficult, due to the largely overlapping distributions of these variables for clear air, pyrometeors and sea clutter. For instance, the pyrometeors / clear air boundary is difficult to define (Fig

1a, 1d). Another example of an overlapping distribution is the co-polar correlation coefficient ($\rho_{HV}$, unitless) values, which tend to produce very similar distributions for pyrometeors and sea clutter, whereby the values of $\rho_{HV}$ show higher frequencies of values in the upper range (above 0.6), with a strong overlap with pyrometeors and sea clutter. Differential reflectivity ($Z_{DR}$, dB) values are more likely negative for sea clutter, but with positive values as well, while the clear air distribution is centred around 0 dB and pyrometeors show more frequency in the higher range, up to 13 dB (the maximum value in the Australian

operational network). The presence of sea clutter is produced by anomalous propagation conditions due to temperature inversion, conditions often present over large water bodies, further enhanced here by the presence of smoke (Guyot et al., 2021). Despite these difficulties, a trained radar scientist could easily differentiate the different echoes from that complex scene. The main challenge here is to automatically discriminate these different echoes, and objectively separate clear air from pyrometeors when these are intertwined.


< Figure 1 here >

While widely used and providing very good results for the segmentation of precipitation and clear air, fuzzy logic and clustering approaches do not make use of the spatial relationship between nearby radar bins (or pixels in the images). In this

paper, we propose a new method for the segmentation of pyrometeors plumes based on the combination of a textural approach (Gray Level Co-occurrence Matrices) and an unsupervised machine learning approach (Gaussian Mixture Model). The paper is organised as follows: we first describe the methods, then evaluate the effectiveness of our new technique using operational weather radar data from several wildfires that occurred in Australia during the 2019/2020 Black Summer.



## 2 Methods

### 2.1 Texture fields based on Gray Level Co-occurrence Matrices

Various methods have been proposed to quantify the texture, i.e., the spatial arrangement of intensities, of an image (Li et al., 2014). First order statistics consist of simply deriving mean or variance of distributions of values within an image. These features can be computed globally, i.e., deriving single values for the whole image, or locally deriving values for each pixel

by applying a moving window and allocating feature values to the centre pixel. These first order statistics only reflect the distribution of intensity levels in an image (Haralick et al., 1973), and do not preserve the directionality of the intensity distribution.

A widely used method in texture analysis proposed by Haralick et al. (1973) relies on the computation of GCLM from which

Haralick features can be calculated. Recent applications include medical image analysis, in particular Magnetic Resonance Sounding (MRI) or ultrasound for the detection of cancers (Chitalia et al. 2019, Yang et al. 2012). However, GLCMs have also been used extensively in the analysis of satellite imagery since it was first proposed by Haralick in 1973 (Hall-Beyer, 2017). The first step in the GLCM approach is to re-scale the original image (with values ranging from $k$ to $l$) to a new quantized image (with integer values ranging from 0 to $N$). This important step can be optimised to reduce the GLCM computational

time, as the smaller the range [0, N], the faster the computation. However, care must be taken if reducing the range past a certain value of $N$ as information contained in the original image will be lost. It must also be noted that different values of $N$ will lead to different values of GLCM and their Haralick features, so the reproducibility of the results strongly depend on that chosen N value. Lofstedt et al. (2019) discussed this issue in detail and proposed a Gray-Level invariant approach to retrieve texture feature values independently of the image quantization. To optimise the computational efficiency of our application, we

decided not to implement this approach.

In a second step, the GLCMs are computed by counting how many times each pair of pixels of the same value (for a given gray level) occurs within a given window surrounding the central pixel. The neighbour of a pixel is defined by a vector of angle $\theta$, and distance $d$. The GLCM can be defined as eq. (1):


$$X(i,j) = \sum_{m=0}^{M} \sum_{n=0}^{N} (1 \; if \; pixel \; identicals, 0 \; otherwise) \tag{1}$$

Where X is the GLCM matrix of elements (i,j) and m,n is the quantized given window. For each displacement vector [combination of (d, $\theta$)] a GLCM can be calculated. To cover all directions surrounding a central pixel, eight angles should be used, but displacement vectors of opposite angles will lead to symmetric GLCMs, therefore only four angles are necessary to



cover all the possible variations (θ = 0, π/4, π/2, 3π/4). We also consider two distances (d=1, d=2), resulting in eight displacement vectors in our study. We discuss the sensitivity of the results to these choices in Section 3.2.

In the third step, the original GLCM is normalised so that all elements of the matrix represent the probability of each combination of pairs of neighbouring pixels to occur for the given window over which the GLCM was calculated. This normalised GLCM is calculated as eq. (2):

$$P_i = \frac{X_i}{\sum_{i=0}^{K-1} X_i} \tag{2}$$

Where $i$ is the pixel number, $P_i$, is the probability recorded for the cell $i$, and $K$ is the total number of pixels.

Finally, in the last step, we computed the Haralick features from the normalised GLCMs. Originally, Haralick et al. (1973) proposed 14 different features to be calculated from the GLCMs. Here, we chose to restrict ourselves to the 6 most used features (and conduct a comparison of these for synthetic and real data to see if these could be reduced further. The 6 chosen features together with their mathematical expressions are shown in Table 1. Each feature can be calculated for each combination of (angle, distance), leading to 8 values for a single feature. A common approach used widely is to take the mean (from 8 values) of each of the features (Lofstedt et al., 2019), this being referred to as a spatially invariant measure (given this is average of the 4 possible directions). We decided to explore the effect of directionality on the retrieved features, by comparing these 8 different retrievals, and we also computed the mean and the standard deviation of the 8 values.

**Table 1:** Selected texture Haralick features utilised in this study together with their mathematical expressions. All the six features are unitless.

| Haralick feature | Expression |
|---|---|
| Contrast | $\sum_{i,j=0}^{N-1} P_{i,j}(i-j)^2$ |
| Correlation | $\sum_{i,j=0}^{N-1} P_{i,j}\left[\frac{(i-\mu_i)(j-\mu_i)}{\sqrt{(\sigma_i^2)(\sigma_j^2)}}\right]$ |
| Homogeneity | $\sum_{i,j=0}^{N-1} \frac{P_{i,j}}{1+(i-j)^2}$ |
| Dissimilarity | $\sum_{i,j=0}^{N-1} P_{i,j}|i-j|$ |
| Angular Second Moment (ASM) | $\sum_{i,j=0}^{N-1} P_{i,j}^2$ |



| Energy | $\sqrt{ASM}$ |
|---|---|

145

## 2.2. The spherical representation of weather radar data

The mode of acquisition of weather radar data, scanning and receiving from the same antenna, and the scanning strategy, dictate that the resulting data are distributed in the three-dimensional space where each grid points can be described in polar coordinates (Elevation, Azimuth, Range). In this article, we perform our calculations on two-dimensional, plan position indicator (PPI) scans in their native spherical coordinates. We are considering position plan indicators as plane (two-dimensional) surfaces that correspond to a given elevation, and its full range of azimuthal and range values. For each of the radar variables, a PPI can be considered a two-dimensional "image" with the x-axis as the Range, and the y-axis as the azimuth. However, the spatial resolution of these images varies considerably along the y-axis because of beam-broadening with range. Typically, for a radar with a beam width of 1 degree, and a range gate size of 250 m, the approximate area covered by a pixel at 1 km Range is 4,909 m$^2$, while the area covered by a pixel at 100 km Range is 10,000 times larger (the area is proportional to the square of the distance). Weather radar data can be gridded using various methods (e.g., Brook et al., 2022), however these methods necessarily smooth the underlying radar fields. Typically, the correlation coefficient or differential reflectivity fields can show boundaries in space when the observed medium includes rain or hail. Clear air, sea clutter and pyrometeors all exhibit very spatially variable signatures (appearing as "noisy") in two-dimensional space. Interpolating native weather radar polarimetric variables on a regular grid, and necessarily smoothing these fields, would impact the retrieved texture, likely reducing its absolute local values and modifying its spatial pattern.

Several authors have employed texture analysis for image classification on weather radar fields. Chandrasekar et al. (2013) reviewed the methods used in classification, including early work on texture. Giuli et al. (1991) and Schuur et al. (2003) applied first order statistics such as mean and standard deviation over a 3x3 window (3 azimuths, 3 ranges) for $Z_{DR}$, $Z_H$ and the differential phase. Gourley et al. (2007) proposed to use the root-mean-square difference between pixels within a 3x3 window, effectively a first order statistic, as it does not consider pixel interdependency, relations in space between pixels of the same values. More recently, Stepanian et al. (2016) and Jatau et al. (2021) utilised variance over a 5x5 pixel neighbourhood for $Z_{DR}$, and the root mean square deviation of differential phase over 9 pixels in Range. All these authors used radar data in polar coordinates; Gourley et al. (2007) noticed the range dependency of texture fields. Their $Z_{DR}$ texture field was decreasing with range close to the radar, where noisy values of $Z_{DR}$ are more common due to clutter. Conversely, the $Z_{DR}$ texture field was increasing in value at long range, due to the natural variability of $Z_{DR}$. This is contradictory to the description of Stepanian et al. (2016) who stated that as the beam width of sampling volumes increase with range, more scatterers are contributing to the volume, thereby increasing the intra-volume variability. Increasing the sample size leads to a more accurate representation of mean values for volume at further range, less pixel-to-pixel variability and therefore smaller values of the texture at further



range. Lakshmanan et al. (2003) used the homogeneity of $Z_H$, where homogeneity is calculated from the co-occurrence of pixel of the same value within a given window (similarly to GLCMs). The authors did not discuss range effects in polar coordinates

and present only succinct results where such effect cannot be observed. Finally, deOliveira and Pereira (2022) explored the application of Gray Level Difference Vector (GLDV) to gridded rainfall data derived from weather radar. The advantage of GLDV over GLCM is significant improvements in computational time. This is because the two-dimensions of the GLCMs are reduced to a single vector of the size of the quantisation. However, their results and interpretations based on gridded data are not directly transferable to polar coordinates.


Here, we propose an adaptation of the GLCM to polar coordinates, by varying the window size along the axis tangential to the Range axis. Indeed, if we consider a given PPI defined by its Range and azimuth in polar coordinates, we can also see the PPI as an image with identical pixel sizes with the range as the x-axis, and the azimuth as the y-axis. As shown previously, the radar volumes and their projected surface areas are varying along the range axis as a function of the square of the range. We

attempt to normalize the neighbourhood area for each GLCM calculation, by including more pixels in the azimuthal direction at close range, and less at far range. The slope of the window size variation shall be like that of the variation of the pixel surface area. We arbitrarily set a minimum window azimuthal width of 5 at long range, since a smaller window would lead to too few pixels to calculate a GLCM. We also set a fixed window range depth of 5, therefore the window at long range has a square shape (in image space). Based on these constraints, we derived a function providing the window width as a function of range.

It should be noted that this approach does not fully account for object-scale effects that are inherent to change of resolution with range. Typically, the radar can resolve objects (such as vortices) at closer range, while suffering from spatial aliasing due to non-uniform beam filling at longer range. Both the fixed window and varying window approaches are implemented on random noise data fields in order to evaluate the benefits of implementing a varying window. That synthetic field in polar coordinates was made of a repetitive pattern of circular shapes of randomly distributed Gaussian noise (Figure 2a), with radius

of approx. 80 km each, so that at least three disks will occur within the 200 km radius of the radar image. The implementation of the above algorithm was done in the Python language (Rossum, 1995) and run using CPUs. A future implementation of our approach using GPUs to improve computational time is discussed in the conclusion section.

## 2.3 Segmentation of data with a Gaussian Mixture Model


Gaussian Mixture Models have demonstrated their efficiency for image segmentation, especially where no pre-conception on the probability distribution of the input features (or variables) are available. This method has also been used to classify hydrometeor types based on weather radar variables (Wen et al., 2015). Here, we employed the same approach as in McCarthy et al. (2019) to classify pyrometeors, i.e., the scatterers typically present in pyrometeors' plumes, using portable X-Band

weather radar data. As Wen et al. (2015) demonstrated, the probabilistic nature of the GMMs can account for any specificity of any given weather radar. It is a more objective approach than fuzzy logic or decision trees approaches, which are based on





pre-conceptions of the data structures. GMM are a modified version of the *k*-means clustering method, and similarly to *k*-means, requires the number of clusters to be chosen by the user (the hyperparameter *k*). The k-means algorithm is improved by using the expectation maximization (EM) method, which was introduced by Dempster et al. (1977). In this algorithm, the

first step (the expectation step) estimates the probability of each data point belonging to one of the k distributions. These distributions are described by their mean and covariance across the input features (in this case, radar variables). In the second step (the maximization step), the mean and covariance of the *k* distributions are recalculated based on the probabilities found in the first step. This results in models that can effectively capture multivariate datasets represented by ellipsoidal confidence functions, which can be used to probabilistically classify new data. The Gaussian distributions also form a generative model,

allowing for the generation of random output samples based on the mean and covariance of the final maximization step.

Here, GMMs is used as our second step for the segmentation of the 2D, PPI field. Based on a sensitivity analysis, where all combinations of variables as inputs were tested (not shown) we retained the following variables as inputs to the GMM: the Haralick local feature "contrast" (Table 1) of the correlation coefficient variable (unitless), the Haralick local feature "contrast"

for ZDR, the Range of the given radar bin (in meters), reflectivity, correlation coefficient, and differential reflectivity. The GMM is used as an unsupervised classifier, so the model is trained and fitted to a dataset of interest, combining all successive PPIs into a single dataset. Two days have been selected to train the model: 29 November 2019, as this day included several occurrences of intertwined clear air, sea clutter and pyrometeors returns, and 2 December 2019, as this day also included scattered showers moving eastwards over the pyrometeors and clear air regions. These two days of data allowed us to include

all type of potential scatterers that could be present in the vicinity of the region, apart from hail, which was infrequent in the Sydney region over the 2019/2020 Black Summer. The model is then applied to data that were not used as part of the training but that were known to contain pyrometeors: the 11 November 2019 was chosen as that day also included the formation of a pyrocumulonimbus, enabling us to evaluate if our method could distinguish the formation of rain droplets within the pyrometeors' plume.


Selecting the number of clusters for the GMM, i.e., an integer value for the hyperparameter *k,* can be done arbitrarily or chosen based on optimisation techniques. The traditional methods to assess the optimal value for *k* are based on minimising the values of Akaike Information Criterion (AIC) or the Bayesian Information Criterion (BIC) (Vrieze, 2012). In practice, the model is trained and tested on the same dataset for a range of *k* values: in our case, we varied *k* from 1 to 10. The values of the AIC and

the BIC should plateau for a threshold value of *k*, making that value the optimum number of clusters (past that number, some clusters will share very similar properties). However, Wen et al. (2015) and McCarthy et al. (2019) both showed that for GMM applied to dual pol radar data, no plateauing was observed possibly due to the very large size of the dataset (McCarthy et al. 2019) or the assumption of mixture of Gaussian distributions for the data. In our study, we know that the minimum number of clusters (value of *k*) should correspond to different types of scatterers that we want to discriminate, namely: clear air, ground

clutter (if some is left after the clutter removal), sea clutter, pyrometeors, and hydrometeors. Using values of *k* above 4 could



also be acceptable, as the same echoes could present different characteristics, for example pyrometeors could present various microphysical properties as in McCarthy et al. (2019), and hydrometeor classification schemes typically include several hydrometeor types to discriminate between rainfall, ice particles (pristine or aggregates or hail), and melting hydrometeors. In our case, if the BIC and AIC present strong declining gradients past $k = 4$, a larger value for $k$ will be retained. The Python-based scikit-learn package (Pedregosa et al., 2011) implementation of GMM was applied.

## 3 Results

### 3.1. Adaptability of GLCM to the spherical representation of weather radar data

To evaluate the effect of the weather radar range bin size on the retrieved GLCM features, synthetic data was utilised as an input to our GLCM algorithm. This is depicted in Figure 2a, where the input data consists of evenly spaced clusters of random noise across the radar grid (polar coordinates), spanning a range of -200 to 200 km in both the x- and y-axes. The values are distributed across the gray scale (0-255). In Figure 2b, a fixed window size of 20 pixels was used to retrieve the contrast. The expectation is that contrast of all pixels of the same size shall be similar to the circle centred at coordinates x = 0, y = 0, since the pixels at very close range for polar coordinates are indeed very similar to one another. As the range increases, pixels become wider (their length stays the same along the azimuthal axis), and the effect of this can be clearly seen in the shape of the nine circles surrounding the central one. The circles at perpendicular azimuths (0, 90, 180, 270) are less distorted than the ones at azimuths (45, 75, 225, 315) because the distances to the origin are smaller, and the rate of increase along the perpendicular axis to the azimuth is larger. This effect is a clear issue as the retrievals will be range-dependent in terms of both the magnitude of the contrast, and its spatial distribution. In Figure 2d, the GLCM contrast was retrieved using a varying window size with range as previously described. The central circle is similar to the fixed window size, as the initial size of the window is the same (win = 20 pixels). The nine circles surrounding the centre have different shapes and magnitudes than the ones shown in Figure 2b. They all exhibit similar contrast values as the central circle, indicating that reducing the window size along the tangential axis to the range has enabled preservation of the shape of the retrievals.

< Figure 2 here >

In Figures 2c and 2e, results for the GLCM correlation and its standard deviation are similar to those for the contrast, as indicated by the preservation of the spatial structure of the synthetic field and the consistent magnitude of peaks across the grid. These results support the moving window approach as a robust approximation to apply GLCM to data in polar coordinates, where the bin (pixel) size increases with range.

### 3.2 Directionality and Haralick features of the GLCMs



A common practice in GLCM calculations is to take the mean of the four main directions (θ = 0, π/4, π/2, 3π/4) to retrieve a
single mean Haralick feature, then described as directionally "invariant". Here, we decided to quantify the variability across
each feature computed out of the GLCM from the 8 combinations of angles (4) and distances (2). The combination of (d =1,
θ = 0) is used as the reference (x-axis), and every other combination (expect one) is evaluated against that reference, plotted
as scatter together with their orthogonal linear regressions. We used orthogonal linear regressions since both variables include
errors. (Kane and Mroch, 2020). In Figure 3 (contrast), actual data collected by an S-Band radar is used as the underlying
dataset. As expected, (θ = π/4) and (θ = 3π/4) show similar results, and a larger value of the distance (d=2) leads to higher
absolute values of the coefficients of the slope of the linear regression than for (d = 1). The most pronounced difference occurs
for (θ = π/2). Overall, the spread of the values is very small and coefficients of the slope of the linear regression vary by less
than 8%, supporting the strategy of using the mean of the 8 combinations. In Figure 4 (correlation), the same underlaying
actual data as for Figure 3 was used. The spread of the data is much larger than for the contrast, indicating that the correlation
is more sensitive to directionality than the contrast.  Nevertheless, coefficients of the slope of the linear regression vary by less
than 12%, supporting here again the use of the mean of the 8 combinations. Based on these results, we decided to also compute
the standard deviation of the Haralick features systematically, to explore the spatial variability of this directional effect.
Although the value of the standard deviation strongly depends on the number of datapoints within a given window.

< Figure 3, 4 & 5 here >

In Figure 5, we explored the relationships between the six chosen Haralick features, by showing density scatterplots across
each Haralick features for the synthetic data. Strong exponential or squared relations exist between contrast, homogeneity,
energy, dissimilarity, and ASM, while the correlation feature does not show any consistent pattern with any of the other
features. This led us to retain only contrast and correlation as other feature will be redundant with the contrast feature.

< Figure 6 here >

To explore the spatial distribution of contrast and correlation for various radar moments ($\rho_{HV}$, $Z_{DR}$ and $Z_H$), we selected the
same event from the 2019/2020 Australian Black Summer bushfires as shown in Figure 1, where pyrometeors, clear air and
sea clutter can be observed within the same PPI. Notably, in Figure 6a, the $\rho_{HV}$ contrast is highest for the sea clutter and the
pyrometeors, showing a strong potential to discriminate these two echoes from clear air using this feature. The standard
deviation of the $\rho_{HV}$ contrast (Figure 6b) is highest in the edges of pyrometeors' plumes and sea clutter: this can be explained
by the edge effects, where a smaller number of pixels are used to derive the GLCM, therefore providing larger directional
variations from one (d, θ) combinations to the next. The $\rho_{HV}$ mean correlation (Figure 6c) is higher for clear air than for the



pyrometeors and sea clutter, providing here again another means of discriminating the two echoes from clear air. Local maxima of ρHV std correlation can also be observed on the edges of the pyrometeors and sea clutter (Figure 6d). Conversely, the values of mean contrast of $Z_{DR}$ are not as high for pyrometeors as in ρHV contrast, and some variability within the pyrometeors plume can be observed with lower values of the $Z_{DR}$ contrast closer to the fire area (Fig. 6e). On the other hand, the contrast of $Z_{DR}$

for sea clutter shows consistent high values apart from the region located the furthest to the North. Both $Z_{DR}$ and ρHV present high values of the mean correlation (Fig. 6g). Here again, standard deviations of contrast and correlation of $Z_{DR}$ show local maxima (Fig. 6f and 6h) illustrating edges of objects, therefore less reliability on the mean features as compared to areas further away from the edges. The retrievals of $Z_H$ contrast and correlation are shown in Fig. 6i-j, but no feature of particular interest was identified in these that could help segment the different echoes.


**3.3 GMM training and labelling**

Two days of S-Band weather radar data from Sydney (Terrey Hills) were selected from the 2019/2020 Black Summer season to build a dataset to train the GMM. These events were chosen so that the data contains a wide variety of clear air, pyrometeors,

sea clutter and precipitation echoes, as discussed previously. For the selected period, this dataset contains over 4,500,000 datapoints. Based on the two criteria (AIC and BIC), an optimum value of $k = 5$ was found, and while AIC and BIC continued to decrease for higher values of $k$, the magnitude of the drop was much lower, and their values plateaued beyond $k = 5$ (not shown). Once trained, the model was applied to these two days of data to qualitatively assess the classification and attribute to each cluster their respective object attribute (pyrometeors, clear air, sea clutter or precipitation).


< Figure 7 here >

In Figure 7a-p, each panel shows a timestamp of radar PPIs with the GMM classified fields at 30 min intervals to assess the temporal continuity of the classification and discuss the interesting temporal evolution of the case. Clear air is distinguishable

around the radar location and within a large radius (green and light blue colours) while three pyrometeors plumes (in red) are visible to the West of the radar, progressively increasing in area as time progresses. The radar bins corresponding to pyrometeors' plumes are labelled as such in the very first frame, showing that only a dozen pixels are sufficient for the clustering algorithm to attribute the label. Increasing areas of sea clutter (in blue) due to increasing anomalous propagation are visible to the East of the radar over the ocean starting in Fig. 7j and covering a half of the ocean within the frame in Fig. 7p.

The labelling here performs well, with only a few points mislabelled as pyrometeors over the ocean for the large sea clutter object. Except in Fig. 7l, 7m and 7p, where the elongated southern pyrometeors' plume is mislabelled at sea clutter in some parts over the ocean. This concerns only a fraction of the total pyrometeors' plume areas, and in Fig. 7p, it is directly adjacent to the sea clutter, therefore a complex scenario for the clustering algorithm to distinguish the two. Fig. 7q provides some insight into each clusters' characteristics, by showing the mean across each variable of the GMM. Clear air (green and light blue)





means correspond to the lowest reflectivity values, lower values of the range, slightly positive values of $Z_{DR}$ (around 2 dB), high values of $\rho_{HV}$ (0.85), and low values of $\rho_{HV}$ contrast. The distinction between the two clusters is on $Z_{DR}$ contrast, with higher values of $Z_{DR}$ contrast for the light blue cluster. Without additional observations, we cannot evaluate that this is due to a range bias or due physical properties of the echoes. The pyrometeors' plume cluster presents on average higher values of $Z_H$ than clear air, usually present at longer range (although as we can see in Fig. 5 this is not necessary), the largest values of $Z_{DR}$

(with a mean at 2.8 dB), relatively low $\rho_{HV}$ (0.83), relatively low $Z_{DR}$ contrast (similar to clear air) but higher $\rho_{HV}$ contrast (mean around 20). Finally, sea clutter shows a strong signature both in $\rho_{HV}$ contrast and $Z_{DR}$ contrast, with higher mean values of $Z_H$ and range, relatively low values of $\rho_{HV}$, and relatively high values of $Z_{DR}$ (2.4 dB).

< Figure 8 here >


To explore the capability of the algorithm to distinguish precipitation echoes, we applied the GMM retrieval to the training day with isolated showers (Fig. 8a-i). On that day, a very thin and elongated pyrometeors' plume is present to the northwest of the radar location, with another smaller plume to the southwest and a smaller pyrometeors' plume to the north of the radar. Scattered precipitation can be observed as a system is moving eastward. Overall, the precipitation object appears correctly

labelled by the GMM, except for isolated pyrometeors' bins within some precipitation cells, as seen in Fig. 8d or 8e. Some complex interactions between clear air, pyrometeors' plumes and precipitation can be observed in Fig. 8g and 8h, and therefore it is difficult here to quantify the performance of the classification. It is likely that some pyrometeors are entrained within the easterly flow, providing cloud condensation nuclei for droplets to form, while the clear air is also disturbed with the showers moving through. Overall, the classification effectively discriminates the major objects that are the main pyrometeors' plumes

from precipitation showers, and clear air. Based on the literature (McCarthy et al., 2018), and the location of the fire source, we can visually discriminate the echoes from the scene to support that validation. The precipitation cluster (Fig. 8q) is characterised by very high $\rho_{HV}$, low $Z_{DR}$ (just above 1), very low $\rho_{HV}$ contrast (as expected since $\rho_{HV}$ is close to unity for precipitation), as well as low $Z_{DR}$ contrast.

< Figure 9 here >

In Fig. 9, the cluster features are presented as joint distributions using kernel density estimation plots with the $\rho_{HV}$ contrast as the x-axis for Fig. 9b-d as this is one of the most discriminant characteristics as we have seen previously from Fig. 9q. As described in the introduction, $Z_{DR}$ and $\rho_{HV}$ clearly overlap for clear air, pyrometeors, and sea clutter, and these variables

cannot solely be used to classify the different echoes. Only the precipitation cluster is well separated from the other clusters with high values of $\rho_{HV}$ and $Z_{DR}$ values just above zero. In Fig. 9b, the $\rho_{HV}$ contrast clearly discriminates clear air from sea clutter and pyrometeors. The probability density estimates show distinct distributions for clear air and sea clutter, while the



pyrometeors distribution spans over a wider range of values. $Z_H$ (Fig. 9d) and range (Fig. 9c) present similar discrimination potential, where clear air at shorter range from radar is often associated with lower values of reflectivity (also an effect of the

radar sensitivity, which is a function of range, so that clear air cannot be observed at long range), while sea clutter is often further away from the radar and essentially shows higher values of $Z_H$. However, this discrimination is not systematic as a strong overlap can be seen across all clusters for both Range and $Z_H$, showing that while these features complement the texture fields, they certainly won't be sufficient to provide an effective classification. Finally, the contrast of $Z_{DR}$ shows some distinction between clear air, pyrometeors, and sea clutter, but with some overlap. The contrast of $Z_{DR}$ only is not

enough to provide a good discriminant but can be used to complement the $\rho_{HV}$ contrast. Overall, the $\rho_{HV}$ contrast appears as the most effective feature to classify the various echo types, with support from the other complementary fields.

**3.4 Evaluation using an independent radar dataset**

While the satisfactory performance of the classifier was demonstrated for cases that were used to train the model, it is necessary to also evaluate the model on data that has not been used for training. The 11 November 2019 was selected since this day featured a pyrometeors' plume moving in the direction of the radar, with consistent clear air echoes through the observation period, and the initiation of a PyroCb cloud later in the day. In Fig. 10a-h, two main pyrometeors' plumes are clearly identified with the largest being in contact with the clear air. While the exact boundary between clear air and pyrometeors cannot be

verified with auxiliary data, it is reasonable to assume that the boundary of the two objects is well defined (clear air, and pyrometeors' plume). At 03:12 UTC (Fig. 10i), precipitation can be seen within the pyrometeors' plume, and this area increases in size in the next four frames. This corresponds to the formation of the PyroCb (as seen in the brightness temperature on Himawari-8, not shown), and precipitation in the lower levels: the PPI shown in Fig. 10 corresponds to the tilt at 0.9 degrees elevation, therefore the northernmost precipitation cluster in figure 10i-l is observed at 110 km from the radar location, e.g., a

corresponding altitude above ground level of 2.2 km. This validation demonstrates the robustness of the method, when trained for days that include a variety of echoes. The possibility to transfer that trained model to radars with other characteristics (beam width, sensitivity, resolution, calibration) will require a dedicated study.

< Figure 10 here >


**4 Discussion and Conclusions**

In this study, we have demonstrated that statistical texture can be retrieved directly from weather radar data in spherical coordinates using an adapted approach based on Gray Level Co-occurrence Matrices. The use of a varying window width

(along the axis perpendicular to the range axis) enabled us to mitigate a range dependent bias in calculated texture fields, an issue documented by other authors (Gourley et al. 2007; Stepanian et al., 2016). We believe that this bias had limited the wider



use of spatial texture to weather radar data to date. For weather radar variables with local spatial variability fields such as $\rho_{HV}$ and $Z_{DR}$, the use of GLCM on interpolated gridded data is not suitable as interpolation will smooth the fields and strongly affect the retrieved texture. Therefore, it is essential to retain information in polar coordinates, further motivating the need for

the varying window width approach. However, our approach only indirectly accounts for relations between resolution of observations and the scale of the observed object, a well-known effect in remote sensing application originally described by Woodcock and Strahler in 1987. For example, vortices that have a typical scale of a few hundred meters can be partially resolved at close distance from the radar but will be lumped at long range along the perpendicular axis to the range axis. Calculating the texture along a much wider window at close range enables us to increase the number of datapoints used to

calculate the GLCM, effectively reducing the effect of local extrema and smoothing the retrieved texture field. The results obtained for synthetic spatially structured random noise show that the varying window approach enables the retention of both the spatial organisation of the fields, and their absolute values. Texture features such as contrast for synthetic data show same minima and maxima at both close and long range, which is not the case when a fixed window is used. Finally, since neither spectrum width nor radial velocity have been used in the classification, these two radar variables can be interpreted

independently to provide insights into the turbulent features of pyrometeors' plumes.

A limitation in validating our echo classification is the lack of a reference classification. We can only qualitatively assess the accuracy of the results based on extra knowledge such as the areas of actively spreading fire, consistency in the time-series, climatological presence, location of sea clutter for the specific radar and diurnal evolution of clear air returns. Based on this

additional information and the dual pol moments, a trained radar expert would be able to manually classify echoes in the complex scene shown in Figure 2 and likely achieve a similar result than our texture and GMM approach. However, a human manual classification would fail to define interfaces between clear air and pyrometeors where the boundaries are blurred. From this perspective, we believe that our results are at least as skilful as those of an expert, and in some cases, likely less biased because of their objectivity. The main advantage of this newly automated classification is that it provides identification of

pyrometeors, providing the foundations to further apply other algorithms. There remain some issues that could see improvements or at least can be flagged with a degree of uncertainty in the retrievals. However, in cases where only few datapoints are available to calculate texture, such as for isolated pixel groups, or the extremities of pyrometeors' plumes, the retrieved texture fields will show large standard deviations across the (distance, angle) combinations. This variability is present due to the small sample size and can result in potentially unrealistic values of the mean texture retrievals. Another ongoing

issue is the mislabelling of datapoints located over land as sea clutter. These datapoints are primarily located within pyrometeors' plumes, and this mislabelling occurs due to the somewhat similar textural properties of pyrometeors and sea clutter. Our correction using a land/sea mask enabled us to address this issue in a straightforward manner. Notably, only a very small number of datapoints are mislabelled as pyrometeors over the sea (in areas where we know these are sea clutter echoes and not pyrometeors echoes). Finally, a greater diversity of echoes in the training dataset could be considered, and the inclusion



of frozen precipitation such as hail and snow echoes would allow for the capture of the full diversity of echoes that can be encountered in the vicinity of the Terrey Hills radar. This would necessitate increasing and optimizing the k-value in the GMM.

We currently see two main limitations to a generalisation and a wider use of our approach. Firstly, texture fields are dependent on several factors such as: radar characteristics, including frequency, resolution, sensitivity, calibration, and accuracy.
Typically, the systematic error of $Z_{DR}$ for pyrometeors is expected to be larger than that for rain or other echoes. The effect of scale as discussed previously and the relation between the observed object and the resolution of the observation will greatly influence textural retrievals. Of particular interest are texture retrievals from portable weather radars observations, (McCarthy et al., 2018) at X- or Ka-Bands, that are being deployed around wildfires and provide unique insights into wildfire plume dynamics and composition. Systematic retrievals of texture fields, in particular the contrast of the correlation coefficient for
such observations, could help interpret these high-resolution datasets by identifying areas or volumes with predominantly similar echoes, or with very large diversities. The absolute values of texture fields are also dependent on the chosen quantisation (rescaling from variable range to a chosen range of gray scale levels). Addressing this issue should be straightforward following the approach proposed by Lofstedt et al. (2019). Secondly, the computational efficiency of texture calculations is a significant issue for operational use. The current implementation requires 3 min of CPU time (on an ARM-based Apple M1 CPU) required
to retrieve texture fields from three radar moments for two Haralick features and for 8 combinations of (distance/angles). This is a well-known limitation of GLCMs (Clausi and Jernigan, 1998), but there are multiple avenues to reduce this computational cost. Initial work exploring the vectorisation of the GLCM implementation, as opposed to nested loops, could reduce computational cost by a factor five, and masking regions with no-data could also drastically reduce the cost by as much as 10 times. Finally, parallelisation of the GLCM using GPUs can reduce the computational cost by several orders of magnitude,
with early testing indicating processing time reduces from 3 min to approximately 10 ms.

The method presented here represents a significant step towards temporal and spatial insights on fire-atmosphere interactions where previously pyrometeor returns have largely been grouped with a broader 'non-meteorological' class of returns. While there has been a number of studies that leverage the highly detailed information available from radar to develop insights on
plume development above wildfires, and even wildfire behaviour, they have been restricted to case study level analyses (McCarthy et al., 2019). This has principally been due to lack of automated assessment of pyrometeor returns, necessitating the manual interpretation and classification, whereas automated hydrometeor classification is well advanced due to a significant body of research. The possibility to automatically assess physical processes, from a statistical point of view, over multi day and multi week fire campaigns, as well as between different fires will be significant for the fire science discipline.
The discussed method will allow temporal examination of fire escalation, area growth and fuel consumption rates as suggested by Duff et al (2017), while being able to be specific about the type (shape, size, permittivity, concentration) of pyrometeors and the presence of deep and moist convection coupled to fires from the radar data alone.



**Code and data availability**

The Australian operational network weather radar data used for this project are available from the NCI catalogue for non-
commercial use at: https://dapds00.nci.org.au/thredds/catalog/rq0/level_1b/catalog.html (Soderholm et al. 2022; last accessed:
20 December 2022). Further, the unprocessed Level 1 (Soderholm et al., 2019) data are also available in the catalogue. The
algorithm described herein is still under development, with two aspects that still require improvement: computational
efficiency, and implementation of an invariant texture feature (Lofstedt et al. 2019). The code described in this paper will be
available on GitHub later in the year 2023.

**Author contributions**

AG developed the code and analysed the data. All co-authors provided regular scientific inputs on the analysis. AG prepared
the original manuscript with contributions from all co-authors.

**Competing interests**

The contact author has declared that neither they nor their co-authors have any competing interests.

**Acknowledgements**

This research is directly supported by *Google.org*, the non-profitable branch of Google. This project was undertaken with the
assistance of resources from the Australian National Computational Infrastructure (NCI) and the Australian Bureau of
Meteorology, both of which are supported by the Australian Government. The open-source libraries Pandas (Pandas
development team, 2010), Numpy (Van der Walt et al., 2011), Scipy (Virtanen et al., 2019), matplotlib (Hunter, 2007), Proplot
(Davis, 2021), Pyart (Helmus and Collis, 2016), netCDF4 (Rew and Davis, 1990), scikit-learn (Pedregosa et al., 2011), in the
Python programming language (Rossum, 1995) were used to develop and implement the code to process the data.

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

**List of Figures**




**Figure 1: Dual pol weather radar fields from the S-Band radar of Terrey Hills, Sydney for the 29 November 2019 at 05:07 UTC:** Spatial and frequency distributions of horizontal reflectivity (a, d), correlation coefficient (b, e) and differential reflectivity (c, f). Red, black, and blue contoured boxes (a, b, c) correspond respectively to pyrometeors, clear air (possibly including remaining ground clutter), and sea clutter echoes. Same colour coding is used in the histograms on subplots d, e, and f. Hotspots (for fire radiative power > 100 MW; acquisition time 4:29 UTC) derived from MODIS (sourced from FIRMS, Giglio et al. 2016) are plotted as red squares on subpanel (a).





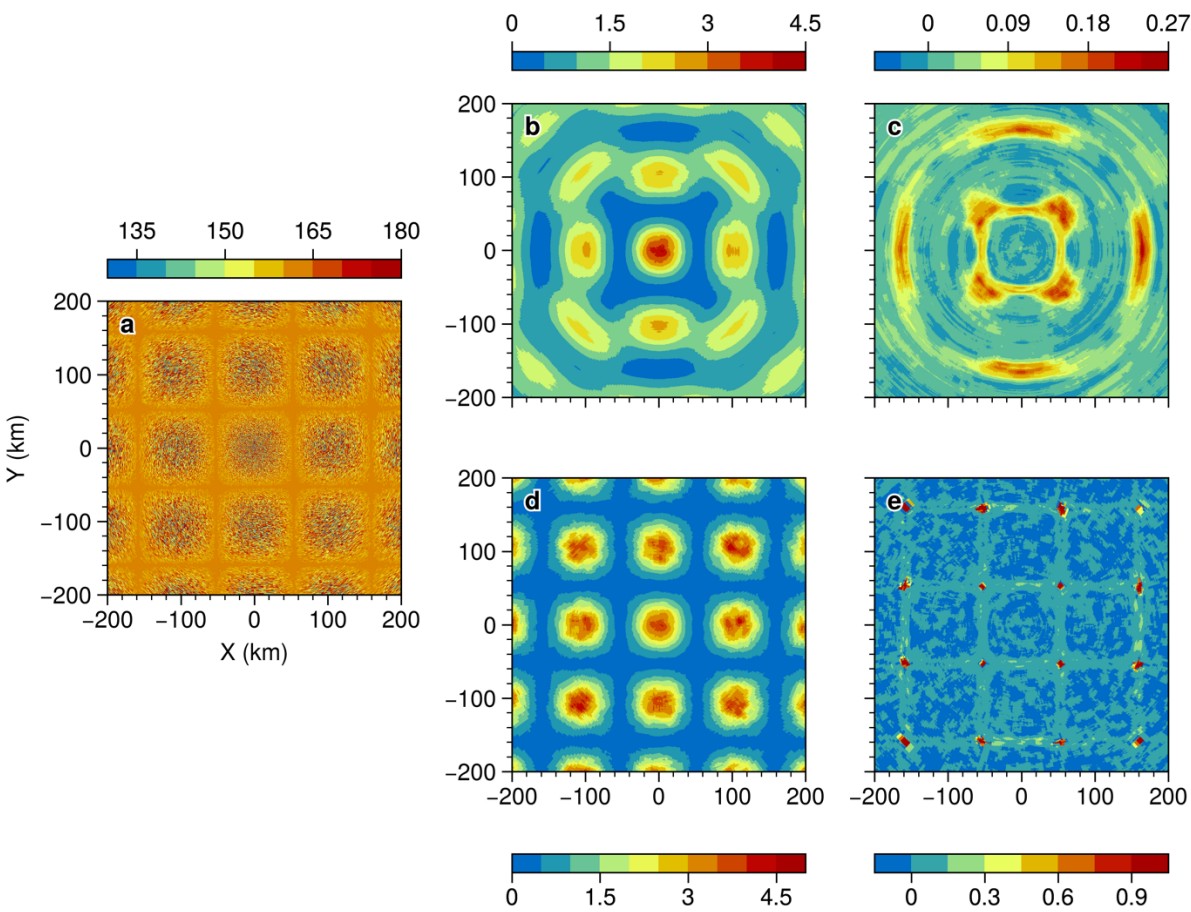

**Figure 2: (a) synthetic field; (b) GLCM mean texture of the synthetic field as shown in (a) with a fixed window size of 20; (c) GLCM mean correlation of the synthetic field with a fixed window size of 20; (d) GLCM mean contrast of the synthetic field with a varying window size; (e) GLCM mean correlation of the synthetic field with a varying window size.**



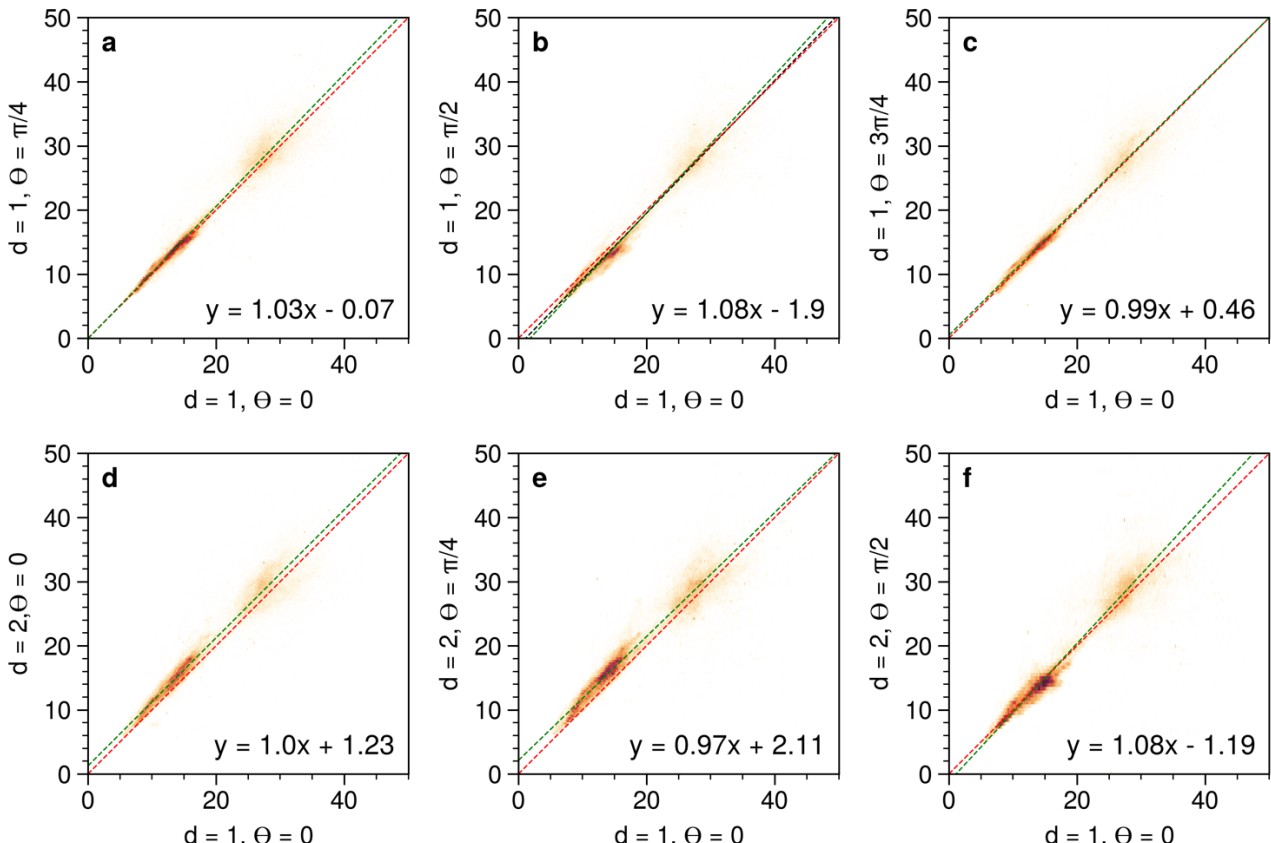

**Figure 3: (a) to (f) scatterplots of GLCM contrast values for different combination of distances (d) and angles (θ) for data from the Terrey Hills radar from the 29 November 2019 at 05:07 UTC for $\rho_{HV}$ at the second tilt (0.9 degrees elevation).**




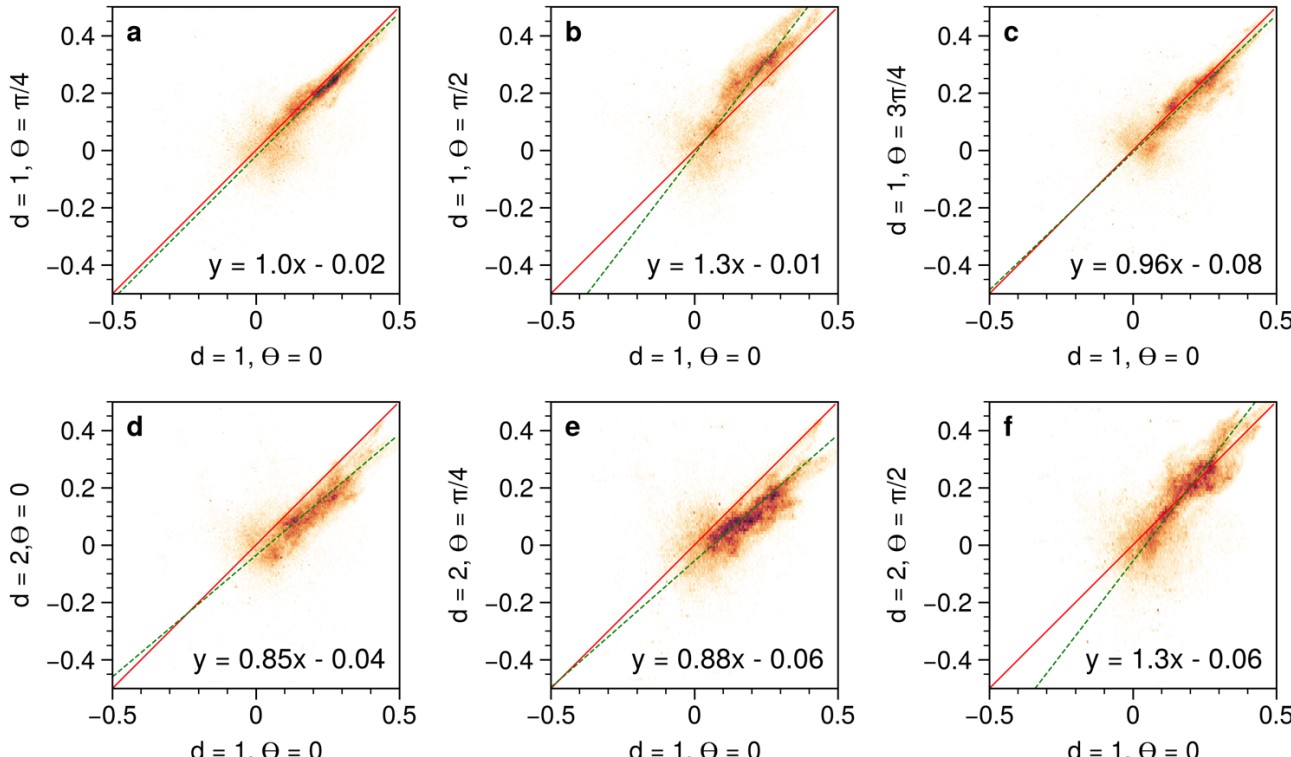

**Figure 4: (a) to (f) scatterplots of GLCM correlation values for different combination of distances (d) and angles (θ) for data from the Terrey Hills radar from the 29 November 2019 at 05:07 UTC for $\rho_{HV}$ at the second tilt (0.9 degrees elevation).**






**Figure 5: Density scatterplots across the selected six mean Haralick features for the structured noise synthetic dataset.**






**Figure 6: Mean and standard deviation of GLCM contrast and correlation calculated from the 8 combinations of angles (θ = 0, π/4, π/2, 3π/4) and distances (d = 1, 2). Data are from the S-Band radar of Terrey Hills (Sydney) for the 29 November 2019 at 05:07 UTC.**





**Figure 7: (a) to (p) timeseries of segmented PPIs (second tilt at 0.9 degrees elevation) from the S-Band Terrey Hills radar for the 29 November 2019; (q) spider plot showing the means across each feature of the Gaussian mixture model. Classified echoes are labelled with the following colour code: gold and lavender for clear air, black for hydrometeors, dark blue for sea clutter and red for pyrometeors. Hotspots (for fire radiative power > 100 MW) derived from MODIS (sourced from FIRMS, Giglio et al. 2016; acquisition time 4:29 UTC) are plotted as dark turquoise.**






**Figure 8: (i) to (p) timeseries of segmented PPIs (second tilt at 0.9 degrees elevation) from the S-Band Terrey Hills radar for the 2 December 2019 including the passage of showers over fire grounds and pyrometeors' plumes. The colour code is the same as for Figure 7. The MODIS overpass prior or concomitant to the radar observations did not detect hotspots, because of total cloud cover over that period.**







**Figure 9: (a) to (d) joint distributions using kernel density estimation showing the distribution of values for each member of the GMM clusters for the training dataset (gold and lavender: clear air, black: hydrometeors, dark blue: sea clutter, red: pyrometeors). A randomly sampled subset (20%) of the training dataset was used for plotting.**







**Figure 10: Timeseries of segmented PPIs (second tilt at 0.9 degrees elevation) from the S-Band Terrey Hills radar for the 22 November 2019. A PyroCb was formed around 03:12 UTC as confirmed by satellite imagery (Himawari-8) of cloud top height. The colour code is the same as for Figure 7-9. Hotspots (for fire radiative power > 100 MW) derived from MODIS (sourced from FIRMS, Giglio et al. 2016; acquisition time 00:06 UTC) are plotted as dark turquoise.**