# Peer review of "Segmentation of polarimetric radar imagery using statistical texture"

_EGUsphere, 2023_

## Author Comment (AC1)

https://egusphere.copernicus.org/preprints/2023/egusphere-2023-181/

**Segmentation of polarimetric radar imagery**

**using statistical texture**

**Authors' response to reviewer's comments**

**9th of August 2023**

We would like to thank the editor and the two reviewers for their very constructive comments on our manuscript. We received genuine insights, which have significantly contributed to increasing the manuscript quality and potential impact. To improve the clarity in our responses we have numbered the reviewers' comments: for example, the comment 1 from reviewer 1 is listed as R1C1 and will refer to these comments as such in the following. We also added a few more information in the manuscript, following comments (as per below) on our preprint by Norman Donaldson (Enviro Canada).

Reviewer #1

The ability to observe large wildfires with proper time and space resolution is mandatory for risk management. Polarimetric weather radars have a chance to identify **pyroclastic** clouds. Nevertheless, microwave signatures of pyroCb are not well distinguished from sea clutter or clear air echoes. The authors propose here a novel approach based on the statistical properties of Gray Level Co-occurrence Matrices (GLCM) and a Gaussian Mixture Model (GMM) to classify echo sources by combining radar variables with texture-based fields. The work is scientifically interesting and the analysis is rigorously conducted and clearly exposed.

Some minor improvements and some further investigations are needed. Section 2.2 deals with weather radar gridded data: to grid those data are needed to move from polar coordinates to Cartesian coordinates. Smoothing these fields is one of the options, usually due to noisy retrieval (e.g. poor sampling), but is not a consequence of gridding. A re-phrase of lines 158-164 is recommended.

R1C1: We disagree that smoothing is not a consequence of gridding. There have been many studies that show common radar gridding methods (e.g. Barnes or Cressman weighted averages) strongly smoothing radar data, including the two mentioned in the line in question (Trapp and Doswell, 2000, Brook et al, 2022). We also note others such as Pauley and Wu (MWR, 1990), Askelson et al (MWR, 2000) and Zhang et al (JTECH, 2005). Please see as an example, the gridded radar product (left), to the polar radar data (right), below from Murillo and Homeyer (JAMC, 2019).

Note how smoothed the gridded product is. One may then imagine how applying the texture technique to the (smoothed) gridded product will arrive at a radically different answer than if it were applied to the original radar data. The following text has been added to clarify this point.

"Weather radar data can be gridded using various methods (e.g., Brook et al., 2022; Trapp and Doswell, 2000); however, these methods necessarily smooth the underlying radar fields and may strongly influence the resulting texture calculations. For this reason, we restrict our texture analysis to data collected in the radar's native spherical coordinates."

[Figure]

In the following lines, the authors mention spatial aliasing: the expression aliasing is commonly referred to wind data from weather radar and not the range of observations. Please consider to re-phrase.

R1C2: A sentence has been added to specify that the intended meaning is general (not only applicable to doppler wind): "Typically, the radar can resolve objects (such as vortices) at closer range but suffers from spatial aliasing (used here its general form, not to be confused with Doppler folding)".

Line 201 "and run CPUs" is not clear. Line 449, frequency, and radar characteristics are indicated as factors influencing texture fields. Please, list factors more specifically detailing the causes of this influence.

R1C3: The sentence was rephrased as: "The implementation of the above algorithm was done in the Python language (Rossum, 1995), and run using multiprocessing and 8 CPUs." At line 452, this text was added to the draft: "Typically, absolute values of radar moments will be affected by the sensitivity, calibration, and accuracy of the radar, and in turn, the spatial field of these moments, and its distributions could be seeing extreme values or outliers, skewed distributions of values, or wider or narrower widths of their distributions. Since the GLCMs are in particular affected by spatial differences between values, an increase or decrease of these differences would affect the texture fields. The radar resolution will also be an important factor affecting the texture because of the averaging of inter-bin variations of radar variables for coarse resolutions, as opposed to a higher bin to bin contrast for higher resolutions. Typically, for various resolutions of the observations and a given size of the observed object (typically vortices of 100s of meters), the resulting variable fields would see lumped values at coarse resolution, where higher resolution would resolve the object, and see strong bin to bin variations. Finally, the radar frequency would also affect the texture field, as for the same scene observed by X- or S- Band radars, one would see more attenuation at X-Band, and different threshold of detection for ash-size particles for example, resulting in different radar variable fields. This frequency effect is though expected to be minimal compared to the ones described above. "

Finally, although the algorithm performance evaluation can not be conducted with direct observations, it could be evaluated as a relative performance with respect to fuzzy logic classification. It is recommended to investigate and discuss this relevant aspect, referring also to the work of Zrnic et al., 2020 (Zrnic, D.; Zhang, P.; Melnikov, V.; Mirkovic, D. Of Fire and Smoke Plumes, Polarimetric Radar Characteristics. Atmosphere **2020**, 11, 363. https://doi.org/10.3390/atmos11040363).

R1C4: As described in the introduction: "Fuzzy logic or unsupervised clustering-based approaches based on polarimetric radar variables are commonly used for weather radar echo classification. For these methods to be effective, each radar echo class needs to occupy a distinct area in the multi-dimensional space defined by all the input parameters with as little overlap as possible, so that robust membership functions can be established (in the case of fuzzy logic approaches) or synthetic models can be developed (in the case of unsupervised learning such as a Gaussian Mixture Model)."

It is inherent to fuzzy logic methods to have an a priori knowledge of what the theoretical distribution of the variables should or could be. Here we use a very different approach, where the data is informing on the properties of the various clusters in the Gaussian Mixture Model. By definition, such data-driven approach can capture what a-priori models cannot capture, and therefore will prove more accurate. This is demonstrated in the validation on unseen data in the paper. We don't think it is necessary to distract the reader of the paper with additional classifications, as the fuzzy logic approach will have the same if not more drawbacks and is not a reference classification to compare to.

References have been added in the introduction section of the manuscript:

"Fuzzy logic or unsupervised clustering-based approaches based on polarimetric radar variables are commonly used for weather radar echo classification (Berenguer et al., 2006; Marzano et al., 2007; Zrnic et al., 2020)."

This text has been added to the discussion section of the manuscript:

"We did not evaluate our approach against other proposed approaches such as Zrnic et al. (2020) as it is well known that by definition, a fuzzy logic-based approach would need a priori information on the distribution of the polarimetric variables, therefore will provide a biased result, and not any reference result to compare with. Zrnic et al. (20 indeed showed that biological echoes (insects/bird) and pyrometeors echoes overlap, and that they observe misclassification with their fuzzy logic algorithm."

**Reviewer #2**

This article does an excellent job of summarizing existing literature related to radar echo classification as it relates to fire-related convection and introduces helpful algorithms for improving the existing methods. This publication should be published after a few minor changes.

Adding more references/justification for not using a cartesian grid would be helpful here (ex. Trapp and Doswell 2000). Including solely Brook et al. 2022 does not provide enough information here. More of a discussion about there the noise comes from in this case (sampling deficiencies) should be discussed. A comparison with a cartesian grid might be a useful exercise here as well.

R2C1: The reference Trapp and Doswell (2000) was added to the reference list. Please see our response in R1C1, where we address this issue while responding to another comment from R1.

More of a discussion and explanation should be adding around line 201 where the author mentions using CPUs vs. GPUs, and why this is important for the computations, possibly including some benchmark estimates or references to related literature on this topic.

R2C2: This paragraph has been extended and now reads: "The implementation of the above algorithm was done in the Python language (Rossum, 1995), and run using multiprocessing and 8 CPUs. Alternatively, GPU-based computing has been shown to be extremely efficient for specific tasks, where parallel computing is possible. Typically, a single GPU can replace dozens to hundreds of CPU cores, as demonstrated by Hafner et al. (2021) for global ocean modelling application, enabling faster and more energy efficient computations. The limitation of using GPU is the possibility of the tasks to be

parallelized, and the need to write the processing code specifically for GPUs. A future implementation of our approach using GPUs to improve computational time is discussed in the conclusion section. "

Häfner, D., Nuterman, R., and Jochum, M.: Fast, cheap, and turbulent—Global ocean modeling with GPU acceleration in Python. *Journal of Advances in Modeling Earth Systems*, 13, e2021MS002717. https://doi.org/10.1029/2021MS002717, 2021.

For the figures (ex. Figures 7 and 8), a legend would help illustrate the different cluster classifications instead of including in the figure description.

R2C3: We have included a new legend on the right side of each figure showing the clusters (new figures 7 to 11).

Also, increasing the quality of Figure 7q.

R2C4: We have separated the original Figure 7q, and created a new Figure 8 to increase the readability of that Figure (and changed subsequently the numbering of original Figures 8 to 10 to 9 to 11). We also slightly change the colour of one cluster (violet) that did not appear well on the previous plot.

Comments from N. Donaldson (Enviro Canada)

One thing the preprint does not explicitly state is the sampling resolution of the underlying radar data used in your analysis. It hints at (250m x 1 deg). Our operational radars collect data in bins of size (500m x 0.5 deg) but the beamwidth is about 0.9 deg so the effective ray width is more like 1.4 deg. A radar with a higher resolution than ours could have more "texture". The number of samples collected on each ray will also affect the texture due to sampling variability.

R3C1: Two sentences have been added to the manuscript at lines 264:" The synthetic data assumes a beam width of 1 degree, and a range gate size of 250 m, similar to the actual data from the Sydney (Terrey Hills) radar later used in the manuscript." and lines332: "The Sydney radar has a beam width of 1 degree, and a gate resolution of 250 m."

Following R1C3, we have extended the discussion on the impact of radar characteristics on the retrieved texture (see R1C3 where we address this aspect). We added a note there on the impact on the number of samples collected on each ray.

Two things that affect measurement of plumes with our radars are ground clutter and biological echoes. We are using quite tall towers so we see more of the ground than many weather radars. Doppler filtering can remove much of that but it leaves holes in areas of weak moving echoes. Another thing I have seen with shallow fire plumes is that biological echoes can dominate over them. Your pre-print only mentions biological signals in passing and in the examples they are spatially separated; are they only a minor issue for your method?

R3C2: The data used in this paper consists of the second tilt of the operational weather radar, which is 0.9 degrees (the first tilt is 0.5 degrees) in order to minimise the impact of ground clutter. Certainly, there would be some clutter that is left despite the data having passed through a QC chain filtering out ground clutter, as explained here: "While a clutter removal algorithm has been implemented to the

data (Gabella and Notarprietro, 2002), some ground clutter is likely remaining due to anomalous propagation conditions occurring on that day, similarly to what was observed in Melbourne under similar conditions (Guyot et al., 2021)."

A sentence has been added in section 3.3 and now reads: "The Sydney radar has a beam width of 1 degree, and a gate resolution of 250 m. All weather data used in this study are from the second tilt at 0.9 degrees elevation, in order to minimise the introduction of ground clutter in the observations, despite its careful removal in the processing chain."

We also added a sentence in the discussion regarding the nixing of biological and pyrometeors' echoes: "There are instances when pyrometeors and biological echoes will mix, and for a given bin, radar echoes would thus be a mixture of both. Because we only used the second (0.9 degree) tilts in this paper, we minimised the potential occurrences of shallow diluted pyrometeors' plumes where biological echoes could dominate over pyrometeors."